# Balancing cardiac privacy with quality in video recordings
Mohamed Elgendi [1,2,3,4] ✉, Aojie Yu[3,4], Saksham Bhutani [1,2,4] & Carlo Menon [3] ✉

## Abstract

**Background** Remote photoplethysmography (rPPG) is a technique that extracts physiological signals, such as heart rate, from facial videos using standard Red-Green-Blue cameras. While rPPG offers valuable health insights, it also exposes individuals to potential misuse, as sensitive information can be inferred without consent.

**Methods** This paper introduces a reversible video modification framework for removing, encrypting, transmitting, and restoring rPPG signals in facial videos, using frame-wise sinusoidal modulation applied to specific rPPG-rich facial regions, with a focus on maintaining perceptual quality and concealing the true heart rate. Our approach is contrasted with prior methods using seven rPPG techniques on the LGI-PPGI dataset, encompassing various activities. Evaluation metrics include PSNR, SSIM, correlation, dynamic time warping, and a composite score reflecting both signal suppression and visual fidelity.

**Results** Here we show that our method achieves an overall score above 0.75 across all rPPG methods, approximately 50% higher than previous approaches. It also demonstrates high visual fidelity (PSNR ≈ 68, SSIM ≈ 0.97) and effectively conceals physiological information, inducing an average heart rate estimation error 22 bpm higher than prior methods.

**Conclusions** This study presents the first end-to-end reversible framework for secure, privacy-preserving video transmission of facial recordings. The approach is lightweight, effective across diverse activities, and holds promise for real-time applications such as video conferencing and telehealth.

## Plain language summary

Facial videos can reveal hidden health information, such as heart rate, by analyzing subtle changes in skin color. While this can help with remote health monitoring, it also poses serious privacy risks, as health data can be extracted without permission. In this study, we propose a method to protect such information. Our approach allows health signals to be removed from a video, encrypted, securely transmitted, and later restored by an authorized recipient. This reversible method keeps the video looking natural while safeguarding sensitive health data against misuse. We test our framework on a set of real videos and show that it works reliably across different activities. This work could improve privacy in video calls, telehealth, and other everyday video applications.

While mobile computing technologies enable remote monitoring of patients' health and easier self-health management, they also raise security and privacy challenges. As defined by the National Committee for Vital and Health Statistics, "Health information privacy is an individual's right to control the acquisition, uses, or disclosures of his or her identifiable health data"[1]. In America, even under the Health Insurance Portability and Accountability Act, which regulates the use and disclosure of Protected Health Information in healthcare treatment, nearly 75% of patients voiced worries about keeping their private health information secure[2]. Most patients reported being "very" or "extremely" worried about discriminatory uses of their personal health information to bar them from insurance, jobs, or healthcare[3].

Remote photoplethysmography (rPPG) is a contactless measurement technique that uses common Red-Green-Blue (RGB) cameras to capture subtle skin color variations in facial videos, thereby monitoring changes in the blood volume pulse[4,5]. With the advancement of computer-vision and signal-processing algorithms, a diverse range of rPPG methods has emerged recently[6–8], enabling robust estimation of vital signs such as heart rate[9], heart rate variability[10], respiration rate[9], oxygen saturation ($SpO_2$)[11], and blood pressure[12]. These developments have powered applications across hospital care[13], telemedicine[14], and fitness training[15], and have facilitated downstream tasks like emotion recognition, stress estimation[16,17], cognitive-load assessment, and personal health monitoring[18]. Recent contributions include a systematic review of rPPG-enabled vital-sign kiosks for chronic and

[1]Department of Biomedical Engineering and Biotechnology, Khalifa University of Science and Technology, Abu Dhabi, United Arab Emirates. [2]Center for Biotechnology, Khalifa University of Science and Technology, Abu Dhabi, United Arab Emirates. [3]Biomedical and Mobile Health Technology Lab, Department of Health Sciences and Technology, ETH Zurich, Zurich, Switzerland. [4]These authors contributed equally: Mohamed Elgendi, Aojie Yu, Saksham Bhutani. ✉e-mail: mohamed.elgendi@ku.ac.ae; carlo.menon@hest.ethz.ch

infectious disease screening[19], identification of the upper medial forehead and malar regions as optimal sites for rPPG acquisition during physical and cognitive activities[20], the proposal of a novel signal-quality index to improve rPPG reliability[21], and a machine-learning framework for reconstructing rPPG signals from standard webcam footage[22].

However, the rise of rPPG techniques has also introduced serious privacy concerns. The presence of physiological signals in facial videos allows for the unauthorized extraction of sensitive health information. Through rPPG, it is possible to infer an individual's physiological state without their knowledge or consent, raising the risk of exploitation in contexts such as targeted marketing, negotiations, or financial transactions[23]. Consequently, there is an urgent need for methods that can selectively modify or obfuscate rPPG-related information in facial videos, ensuring the protection of vital signs while preserving overall video fidelity. In this work, we particularly focus on the removal of physiological signals from facial videos transmitted in scenarios such as video calls, live streaming, and remote patient monitoring, ensuring privacy protection without compromising the visual quality of the transmitted video.

Previous studies[23–27] have proposed various methods for editing rPPG-related information in facial videos. However, these approaches often suffer from major drawbacks, such as time-consuming processing steps, including optimization techniques and principal component analysis, or the introduction of visible artifacts. As a result, they are unable to meet the demands of real-time performance necessary to prevent unauthorized physiological monitoring during online meetings or video calls. While these methods aim to modify the intrinsic rPPG signals in facial regions, none of the previous work directly evaluates the similarity between the rPPG signal extracted from the modified video and the ground truth PPG signal. Instead, they focus on heart rate as a derivative metric for evaluation. Furthermore, video fidelity in these methods is typically assessed using peak-signal-to-noise ratio (PSNR) and structural similarity index measure (SSIM), but these metrics may not fully capture the nuances of signal modification effectiveness.

To address the limitations of previous methods, we propose a framework that includes a sender who modifies the video to obscure the rPPG signals and a receiver who decrypts and recovers the original video using a secure key. To the best of our knowledge, this is the first reversible method that allows the physiological signals to be reintroduced into the video. In our approach, we generate a list of frequencies using RSA encryption. A sinusoidal signal, based on these frequencies, is added frame-wise to the green channel of the video's time series. Along with the modified video, the encrypted frequency list is transmitted to the receiver, who can then decrypt the list and reverse the modification, restoring the original physiological signal. This setup ensures that anyone with access to the video cannot view or extract the original rPPG information, nor can they discern how the video was altered, as the frequency list remains securely encrypted.

Our method represents the first end-to-end reversible approach for rPPG signal removal and restoration that operates frame-wise without explicitly extracting the underlying signal, thus simplifying the overall pipeline. We evaluate the effectiveness of this approach using a tailored metric that compares rPPG waveforms through correlation and dynamic time warping (DTW), and propose a combined score that reflects both signal suppression and video quality. The results show that our method generalizes well across various activities and preserves video fidelity, offering a practical solution for secure and privacy-preserving transmission of physiological video data.

## Methods
### Previous methods
This section provides a review of previous methods. We can categorize prior algorithms for modifying rPPG signals in videos into two groups: deep learning and non-deep learning. The deep learning ones aim to train a neural network to either directly inject or inject the rPPG signal after the removal of rPPG information. Non-deep learning methods rely much on optimization to remove the rPPG-related statistics from video. Our method

is distinguished by its simplicity and lightweight design, applying a frame-wise sinusoidal modulation to the green channel to effectively obscure rPPG signals without the need for complex optimization or training processes. Unlike other approaches that require the estimation of a time-series signal from multiple frames, our method operates on a single frame, eliminating the dependency on temporal information and reducing computational complexity.

For comparison, we selected two representative methods from the literature. First, we re-implemented Chen et al.'s method[25], which removes rPPG signals by applying a soft elimination factor to the principal component with the largest normalized band power. This method serves as an example of a classical signal-processing approach. Second, we attempted to re-implement Sun et al.'s method[26], which relies on a pre-trained 3DCNN-based optimization framework, similar to a projected gradient descent (PGD) attack. However, Sun et al.'s 3DCNN model and training details were not publicly available. Due to this, we used EfficientPhys[28], a pre-trained model on the PURE dataset[29], as a substitute for Sun et al.'s 3DCNN. While this allowed us to compare deep learning approaches, we acknowledge that this substitution may not fully reflect the performance of Sun et al.'s original implementation.

### Our method
Our method includes a module for video modification, enabling secure video transfers without leaking rPPG information to address privacy concerns. We build upon our previously developed framework, rPPG-Removal[30], to implement the video-editing pipeline proposed in this work.

The overview of our framework is shown in Fig. 1. For the secure transfer of video, we first apply RSA encryption, where both the sender and receiver possess a pair of public and private keys. The sender publishes the former to everyone, while the receiver keeps the latter to themselves. The sender creates a list of modified frequencies, such as [100, 150,…], which indicates the use of a bpm 100 signal for the first $t$ seconds and a bpm 150 signal for the subsequent $t$ seconds. RSA can also generate the list from the file name, ensuring the randomness of the inserted heartbeat signal. The receiver receives an encrypted version of the frequency list along with the video, which they will later decrypt to reverse the modifications made by the sender. The encryption of the frequency list prevents anyone who owns the video from seeing the rPPG information in the original video and from understanding the modifications made to it.

We provide a method for generating a random frequency list within the normal heart rate range using RSA. The sender can specify a string that he/she wants to use to generate a frequency list. And then the sender encrypts the string with his/her public key to get an RSA-encrypted string of 512 bytes. Next, we identify the decimal corresponding to each byte, modify it by 101, and then add 60 to convert it into a number within the normal heart rate range (60–160). This process generates a list of 512 random frequencies that fall within the normal heart rate range. Real implementation of the RSA algorithm includes randomized padding during encryption. This ensures that even if the user tries to generate frequency lists with the same string and public key, each time the encrypted string is different and thus produces a different frequency list.

We use the RSA algorithm for encryption the frequency lists. The foundation of RSA lies in the difficulty of factoring large integers. Each user has a pair of keys: a public key to be distributed to anyone who wants to encrypt and send a message to the user $(N, e)$, private key to be kept secretive by the user for decrypting messages $(d, e)$. Two distinct large prime numbers $p$, $q$ are chosen, and the modulus $N$ is computed as

$$N = p \times q \qquad (1)$$

The public exponent is chosen as a co-prime of $(p-1)(q-1)$, that is

$$gcd(e, (p-1)(q-1)) = 1 \text{ and } 1 < e < (p-1)(q-1) \qquad (2)$$

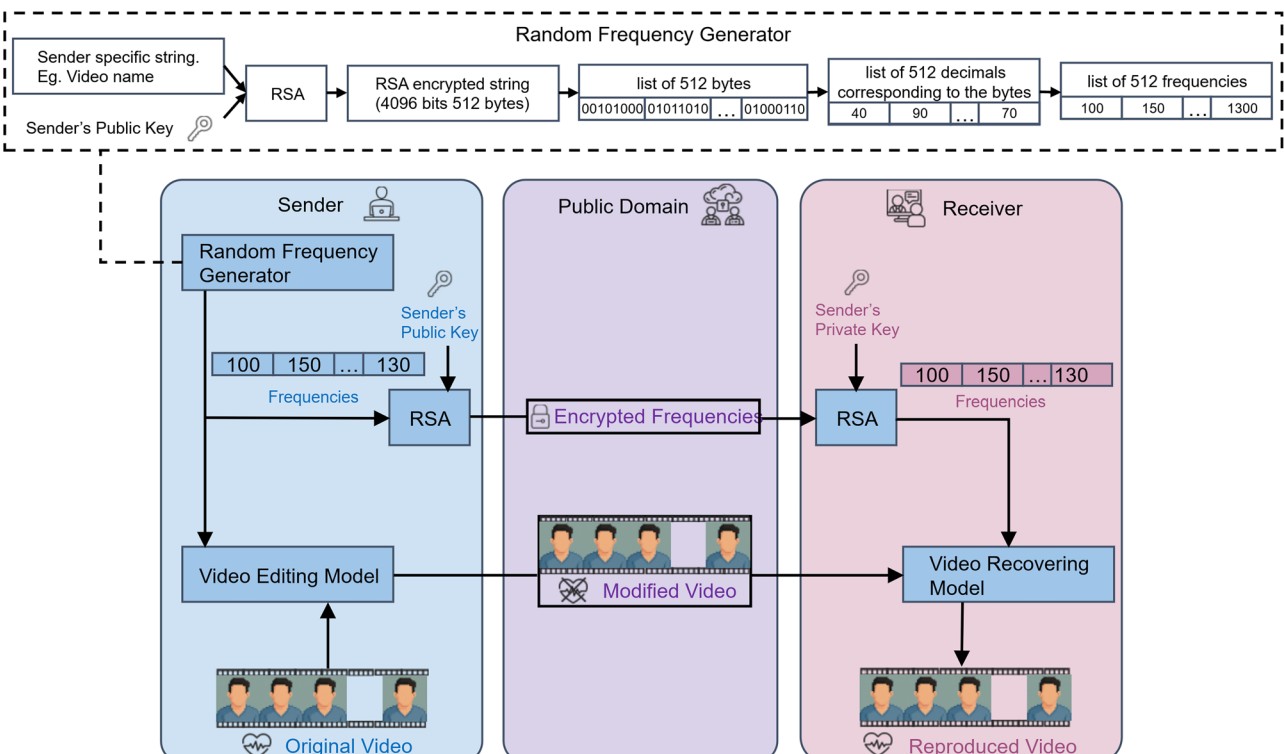

**Fig. 1 | Overview of our framework.** In this setup, a sender transmits a heartbeat-concealed facial video securely to a receiver. The sender begins with the original video and a list of frequencies to obscure the rPPG signal. Using the video-editing model, the sender modifies the original video to conceal the rPPG information, then encrypts the frequency list with the receiver's public key via RSA. Both the modified video and the encrypted frequency list are sent to the receiver, ensuring that only the receiver, with their private key, can decrypt the list, reconstruct the original video, and retrieve the rPPG signal. This process protects the original physiological information from unauthorized access. The figure includes a Random Frequency Generator, which generates a frequency list within a typical heartbeat range for secure modulation.

The private exponent is computed as

$$d = e^{-1} \bmod (p-1)(q-1) \tag{3}$$

A given message $m$ is encrypted through

$$M = m^e \bmod N \tag{4}$$

where $M$ is the encrypted message. And the encrypted message can be decrypted by the user through

$$m = M^d \bmod N \tag{5}$$

If one wants to attack the RSA encryption, one needs to factorize the large number $N$ into $q$ and $p$, which is not possible with a normal computer (but could be possible for a quantum computer), for $N$ as large as 512 bytes.

We notice from rPPG detection methods that not every pixel contains rPPG information: only the face region, specifically the upper medial forehead, lower medial forehead, glabella, right malar, and left malar, conveys the most similar signal to the ground truth rPPG. Moreover, multiple studies[31,32] have shown that the green channel carries the most rPPG information, owing to hemoglobin's peak absorption at green wavelengths. While the red and blue channels contain minor physiological information, they are less reliable for heart-rate extraction and contribute more noise. Thus, we focus our modifications solely on the green channel to maximize the masking of rPPG information while minimizing video distortion. Limiting the modification to the green channel also ensures lower computational overhead. Inspired by these findings, we inject a sinusoidal signal into the green channel over time by editing the five key facial regions. Figure 2 illustrates our video-editing pipeline.

Our method has two main steps and is performed frame-wise. For each frame, we first detect the largest face with MediaPipe Face Mesh and create a binary mask to mask our target area for modification. Next, we directly increase the green channel value by a small amount, adhering to this equation:

$$G(x,y,n) \leftarrow G(x,y,n) + A\sin(2\pi f_{Hz} t_n), \quad t_n = \frac{n}{\text{fps}}. \tag{6}$$

with

| | | |
|---|---|---|
| $G(x,y,n)$ | : | green-channel intensity at pixel $(x,y)$ in frame $n$, |
| $A(=2)$ | : | sinusoid amplitude(grey-level units), |
| $f_{Hz}$ | : | fake-heartbeat frequency(Hz), |
| fps | : | video frame rate(frames s$^{-1}$), |
| $t_n$ | : | elapsed time at frame $n$ (s). |

An example of the rPPG signal estimated from the modified video is shown in Fig. 3. As illustrated in Fig. 4, the modified video introduces additional frequency components around the injected 100 bpm signal, confirming the successful perturbation of the original rPPG signal.

**Video recovering model.** Using the lossless video coding format FFV1 ensures that we can recover the original video from the modified video by inverting the modification process. If we use a lossy video coding format such as MPEG4, when we try to recover the original video from the modified video by inverting the modification process, the traces of modification will still remain, and the original heart rate cannot be recovered exactly.

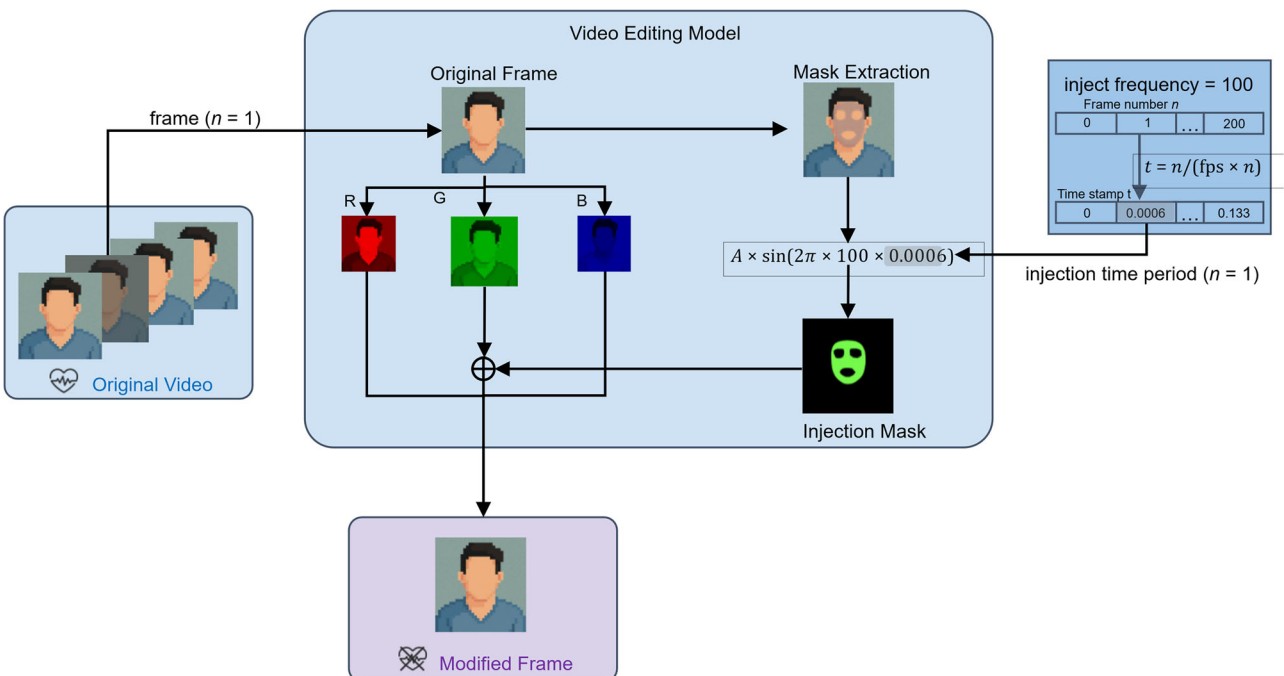

**Fig. 2 | Schematic of our video-editing model.** Starting from an original frame, we extract masks over five facial regions, generate a time-varying injection mask (sinusoid at 100 Hz scaled by amplitude $A$), apply it only to the green channel, and then recombine all channels to produce the modified frame.

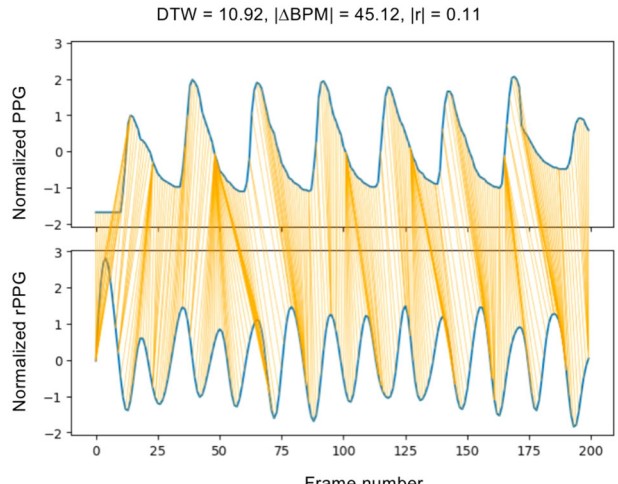

**Fig. 3 | Comparison of reference PPG and modified rPPG signals for a time window.** Normalized ground truth PPG (top panel, blue line) and rPPG signal extracted from the modified video (bottom panel, blue line) using the CHROM method ($n = 1$) are shown for the `cpi` participant ($n = 1$), `rotation` activity ($n = 1$), an 8-s window selected from the full video in the LGI-PPGI-DB dataset. Orange lines indicate the dynamic time warping (DTW) alignment path between the two signals, illustrating temporal correspondence. The PPG and rPPG signals are normalized. Reported metrics are: DTW distance = 10.92, absolute heart rate difference $|\Delta BPM| = 45.12$, and absolute correlation $|r| = 0.11$.

## Dataset description and ethical compliance

For the evaluation, we used the LGI-PPGI dataset from Pilz et al.[33]. This dataset is published under the CC-BY-4.0 license. The study was supported by the German Federal Ministry of Education and Research (BMBF) under the grant agreement VIVID 01|S15024 and by CanControls GmbH, Aachen. The LGI-PPGI dataset contains videos from six participants: one female participant and five male participants. Each participant performed four different activities, each recorded as a 1–5-min video under varying illumination conditions. In the Resting activity, the participant was seated still with no head or facial motion under indoor static illumination. In the Rotation activity, the participant rotated their head left and right under indoor static illumination. In the Gym activity, the participant exercised on a bivalve ergometer under indoor static illumination. In the Talk activity, the participant engaged in an outdoor urban conversation under natural, varying illumination.

Each video is paired with a reference ground truth PPG signal measured with a CMS50E PPG device, a finger pulse oximeter, and a 60 Hz average sampling rate. The videos of resolution 640 × 480 are recorded with a Logitech HD C270 webcam with a 25 Hz average sampling rate.

### Ethical considerations for secondary use

The dataset used in this study was collected and released by the original authors with the requisite ethical approvals and informed consents, in accordance with the regulations of their institution. Responsibility for managing ethical compliance rested with the original data providers, who ensured that the dataset could be distributed and reused under open licensing terms with appropriate acknowledgment.

As this study relies exclusively on secondary analysis of a de-identified, publicly available dataset, no additional Institutional Review Board approval or informed consent was required. This practice is consistent with widely accepted research standards for the use of anonymized, open-access data.

A detailed description of the LGI-PPGI dataset, including its source and licensing terms, is provided in the manuscript to ensure transparency and adherence to both ethical and legal requirements.

### Evaluation pipeline

We evaluate rPPG modification algorithms from two key perspectives: the effectiveness of removing the original rPPG-related information and the resultant video fidelity. Instead of limiting our analysis to derivatives like heart rate, we perform a more comprehensive evaluation by directly comparing the rPPG signals. This approach captures the broader impact of signal modification, as rPPG contains rich physiological information beyond heart rate alone. By considering both signal accuracy and video quality, our evaluation provides a more thorough understanding of the trade-offs between privacy protection and visual fidelity. Figure 5 shows the evaluation pipeline.

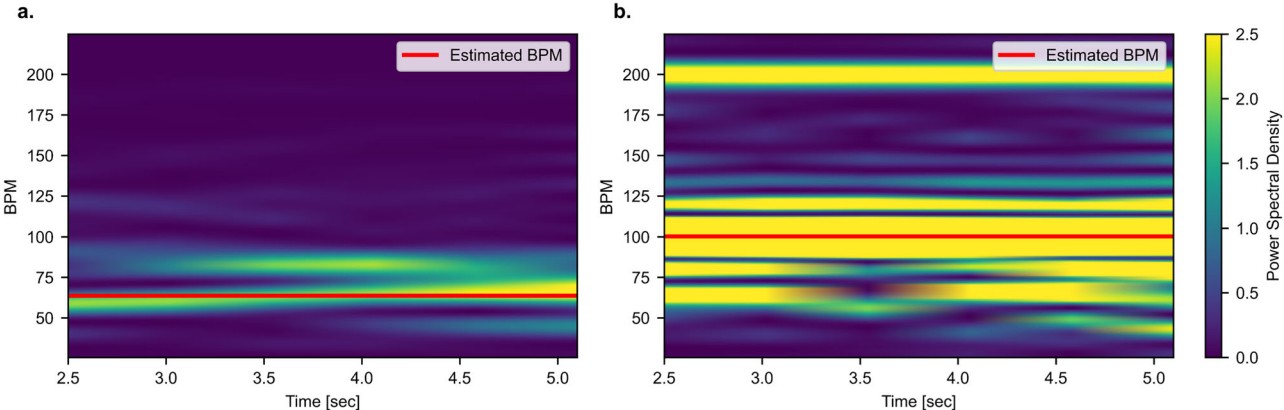

**Fig. 4 | Spectrogram comparison of rPPG signals. a** The unmodified video and (b) the modified video with an injected frequency of 100 bpm and amplitude $A = 2$. Both spectrograms are derived from a selected time window of the same video segment, recorded using the CHROM rPPG method ($n = 1$) for the `cpi` participant ($n = 1$), `rotation` activity ($n = 1$), an 8-s window selected from the full video in the LGI-PPGI-DB dataset. The red line represents the estimated BPM for the time window. In

(**a**), the original spectrogram shows a dominant frequency component closely aligned with the estimated BPM, indicating the presence of the true physiological signal. In (**b**), after modification, additional frequency components appear in the spectrogram, especially around the injected frequency of 100 bpm, indicating successful perturbation of the original rPPG signal.

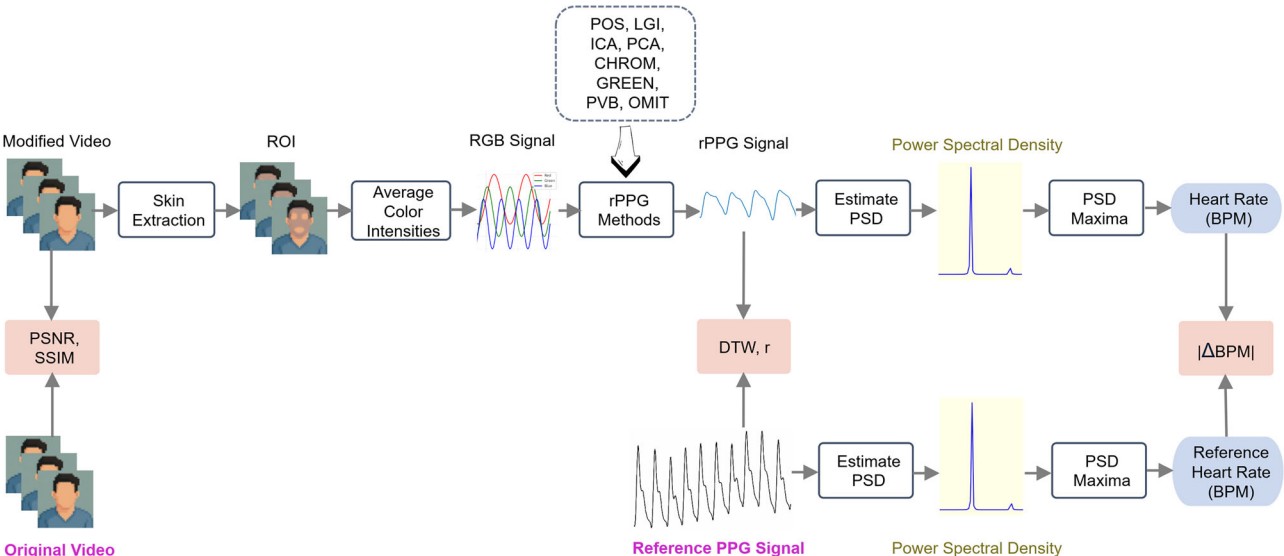

**Fig. 5 | An illustration of the pipeline for evaluating the performance of rPPG modification methods.** This schematic illustrates the process for evaluating the impact of rPPG modification methods on both signal integrity and video quality. The modified and original videos are processed to extract skin regions and compute average RGB color intensities from the region of interest (ROI). These RGB signals are then passed through a set of rPPG algorithms (POS, LGI, ICA, PCA, CHROM, GREEN, PVB, OMIT) to estimate the rPPG signal. The signal is analyzed in the

frequency domain via power spectral density (PSD) estimation, and the heart rate is derived from the dominant frequency. Comparison with the reference PPG signal allows computation of signal-level metrics such as DTW, $|r|$, and $|\Delta \text{BPM}|$. Additionally, video quality metrics like peak-signal-to-noise ratio (PSNR) and structural similarity index measure (SSIM) are computed between the original and modified videos to quantify visual distortion.

For video fidelity, we consider two measurements: PSNR and SSIM.

*PSNR* is a full-reference video quality measurement based on byte-by-byte data comparison[34]. It calculates the ratio between the highest attainable signal power and the power of the interfering noise[35]. For fixed content, PSNR correlates highly with subjective perceptual quality[36].

$$\text{PSNR} = 10 \log_{10}\left(\frac{P_{\max}^2}{\text{MSE}}\right), \qquad (7)$$

$$\text{MSE} = \frac{1}{M \times N}\sum_{x=1}^{M}\sum_{y=1}^{N}(I(x,y) - I'(x,y))^2, \qquad (8)$$

where $I(x, y)$ and $I'(x, y)$ are the original and modified frame intensities at pixel $(x, y)$, $M$ and $N$ are the frame width and height in pixels, $P_{\max}$ is the maximum possible pixel value, and $\log_{10}$ denotes the decimal logarithm (base 10).

*SSIM* is a full-reference image quality metric that measures the perceptual similarity between a modified image and its original. It evaluates three components–luminance, contrast, and structure–each of which models an aspect of human vision[37]. Based on the assumption that the human visual system is highly sensitive to structural information, SSIM is defined as

$$\text{SSIM}(\mathbf{x}, \mathbf{y}) = [l(\mathbf{x}, \mathbf{y})]^\alpha \times [c(\mathbf{x}, \mathbf{y})]^\beta \times [s(\mathbf{x}, \mathbf{y})]^\gamma \qquad (9)$$

where,

$$l(\mathbf{x}, \mathbf{y}) = \frac{2\,\mu_x\,\mu_y + C_1}{\mu_x^2 + \mu_y^2 + C_1} \tag{10}$$

$$c(\mathbf{x}, \mathbf{y}) = \frac{2\,\sigma_x\,\sigma_y + C_2}{\sigma_x^2 + \sigma_y^2 + C_2} \tag{11}$$

$$s(\mathbf{x}, \mathbf{y}) = \frac{\sigma_{xy} + C_3}{\sigma_x\,\sigma_y + C_3} \tag{12}$$

where: $\mathbf{x}$ and $\mathbf{y}$ are the original and modified video frames, $l$, $c$, and $s$ compare luminance, contrast, and structure respectively, $\mu_x$, $\mu_y$ are the mean pixel intensities of $\mathbf{x}$ and $\mathbf{y}$, $\sigma_x^2$, $\sigma_y^2$ are their variances, and $\sigma_{xy}$ is their covariance, $\alpha$, $\beta$, $\gamma > 0$ weight the relative contributions of each component, and $C_1$, $C_2$, $C_3$ are small constants to avoid instability.

We validate the effectiveness of the proposed method by calculating the correlation coefficient and DTW between the modified rPPG signal and the original PPG signal. These metrics assess both temporal alignment and signal similarity, which are critical to evaluating whether the true rPPG signal has been successfully removed or masked. Many studies have shown that metrics such as the correlation coefficient[38] and DTW[32,39] are effective in comparing PPG signals. A substantial reduction in these metrics indicates successful removal or masking of the true rPPG signal.

*DTW* is an efficient measurement for time-series similarity. It lets you use elastic transformation to find the best way to line up time-varying sequences while staying within certain limits. It finds shapes that are similar while reducing the effects of time shift caused by speed difference and distortion[40,41]. The data is transformed by multi-scaling and resampling to get the alignment path, and the Euclidean distance between the points is computed. The DTW is computed between the normalized rPPG and ground truth PPG signal for each window.

*Correlation coefficient r* is computed between the normalized rPPG and the ground truth PPG signal for each window.

$$r = \frac{\sum_{i=1}^{n}\left(x_i - \frac{1}{n}\sum_{j=1}^{n} x_j\right)\left(y_i - \frac{1}{n}\sum_{j=1}^{n} y_j\right)}{\sqrt{\sum_{i=1}^{n}\left(x_i - \frac{1}{n}\sum_{j=1}^{n} x_j\right)^2 \sum_{i=1}^{n}\left(y_i - \frac{1}{n}\sum_{j=1}^{n} y_j\right)^2}} \tag{13}$$

where $x_i$, $y_i$ are points of the PPG signal and points of the rPPG signal.

*Beats-per-minute difference* ($\Delta BPM$) is the mean absolute difference between the estimated heart rate and the reference heart rate.

*Overall score* is proposed for easy and straightforward comparison of the relative performance of different methods in editing rPPG signals while preserving video quality. The overall evaluation score was calculated as follows:

$$\text{OS} = \frac{1}{4}(\text{DTW} + (1 - r) + \text{PSNR} + \text{SSIM}), \tag{14}$$

where DTW and $r$ are the min-max normalized DTW distance and Pearson correlation coefficient between the original and modified videos, PSNR and SSIM are the min-max normalized PSNR and SSIM of the modified videos, and OS is the overall evaluation score ($0 \le \text{OS} \le 1$).

## Implementation details

For comparison, we re-implemented the newest method from each rPPG method category. We re-implement Chen's method[25] according to the description in Chen's paper. We also try to re-implement Sun et al.'s method[26]. However, the pre-trained 3DCNN Sun et al. used is not open-sourced, and the details about the 3DCNN and the training process are not disclosed. We followed Sun et al.'s method[26] but used EfficientPhys[28] pre-

trained on PURE[29] dataset instead. We are aware that Sun et al.'s method's[26] performance is highly related to the performance of the pre-trained neural network, and our choice may not accurately reflect Sun et al.'s[26] actual implementation.

For double-cycle consistency learning, they released the model but did not release the training parameters. We need to train the model. In the paper, the model undergoes training on a large dataset, UBFC, which we cannot access due to insufficient GPU memory. Instead, we trained the model on PURE, a process that took 3 days. The rPPG-removed videos exhibit a noticeably lighter color compared to the original footage. We observed large color blocks in the output videos when the subjects rotated their heads. Therefore, we have decided to exclude this method from the final evaluation.

The code developed and used in this study is publicly available at Zenodo[42]. MediaPipe Face Mesh[43] was used to detect landmarks that enclose the region of interest (ROI) on a participant's face for each video frame. We use a holistic approach to get a single patch of the PPG-related facial area, the whole facial skin region as a convex hull of 468 facial landmarks, except those related to the eyes and mouth. Then, a single estimator for the RGB signal can be derived by averaging the facial skin color intensities for each channel within the ROI.

The RGB signal is converted to an rPPG signal using an rPPG method. In our experiments, we tried eight different rPPG methods: GREEN[31], ICA (Independent Component Analysis)[44], PCA (Principal Component Analysis)[45], CHROM (Chrominance-based method)[46], PBV[47], POS (Plane-Orthogonal-to-Skin)[48], LGI (Local Group Invariance)[33], OMIT (Orthogonal Matrix Image Transformation)[49]. Then a sixth-order Butterworth band-pass filter is applied to the rPPG signal to only keep this within the range 0.65–4.0 Hz, which is equivalent to 39–240 bpm. For the rPPG methods, we follow the implementation from the package pyVHR (Python Framework for Virtual Heart Rate)[38].

The estimated rPPG signal and the reference ground truth fingertip PPG signal are both split into 8-s overlapping windows, corresponding to 200 frames in the LGI-PPGI dataset.

For each window, we use Z-score normalization to normalize the rPPG signal and the reference PPG signal.

Given the estimated rPPG signal, the associated beats-per-minute is derived through estimating the corresponding power spectral density (PSD) of the rPPG signal within each specified window using Welch's method, and the PSD maxima represent the instantaneous heart rate.

## Reporting summary

Further information on research design is available in the Nature Portfolio Reporting Summary linked to this article.

## Result

Table 1 presents the evaluation metrics for our method, Chen's method[25], and the PGD attack[26], separated by four different activities—gym, resting, rotation, and talking—as well as different rPPG measurement techniques.

The PSNR column in Table 1 shows the average PSNR between the modified video and the original video across activities. Our method consistently achieves substantially higher PSNR values compared to both Chen's method and PGD, approximately doubling their performance across all activities. Specifically, our method reaches an average PSNR of 67.87, compared to 37.02 for Chen and 33.48 for PGD. In employing our method, we utilize the lossless video coding format, FFV1. Consequently, the frames wherein $sin(0)$ is added to the green channel remain identical to the original frames, causing the PSNR to approach infinity. To facilitate numerical comparison, we have capped the maximum PSNR at 100 instead of allowing it to reach infinity.

As shown in Table 1, our method consistently outperforms PGD by approximately 0.1 SSIM across all activities. The difference between our method and Chen's method is negligible, with both achieving an average SSIM of 0.97. Notably, Chen's method performs slightly better in the gym

**Table 1 | Value of the average DTW, |r|, |ΔBPM| over all participants, and windows for every rPPG method on original video, video modified by our method, by Chen's method[25], and by projected gradient descent (PGD) attack[26] grouped by video activity**

| Video activity | rPPG method | |r|↓ | | | | DTW↑ | | | | |BPM| (MAE)↑ | | | | SSIM↑ | | | PSNR↑ | | |
|---|---|---|---|---|---|---|---|---|---|---|---|---|---|---|---|---|---|---|---|
| | | Original | Our | Chen's | PGD | Original | Our | Chen's | PGD | Original | Our | Chen's | PGD | Our | Chen's | PGD | Our | Chen's | PGD |
| Gym | CHROM | 0.22 | 0.13 | 0.15 | 0.14 | 7.63 | 6.95 | 8.05 | 7.28 | 24.79 | 26.34 | 37.64 | 23.94 | 0.94 | 0.97 | 0.81 | 60.97 | 36.71 | 32.21 |
| | GREEN | 0.24 | 0.14 | 0.17 | 0.15 | 7.88 | 7.22 | 8.34 | 7.59 | 34.32 | 27.73 | 40.25 | 32.91 | | | | | | |
| | ICA | 0.23 | 0.14 | 0.17 | 0.14 | 7.9 | 7.32 | 8.34 | 7.6 | 31.41 | 37.2 | 40.07 | 34.11 | | | | | | |
| | LGI | 0.25 | 0.14 | 0.17 | 0.14 | 7.72 | 7.15 | 8.24 | 7.45 | 20.21 | 26.21 | 35.19 | 23.32 | | | | | | |
| | OMIT | 0.22 | 0.14 | 0.16 | 0.14 | 7.84 | 7.29 | 8.27 | 7.54 | 20.22 | 26.28 | 35.12 | 23.13 | | | | | | |
| | PBV | 0.22 | 0.14 | 0.16 | 0.14 | 7.87 | 7.32 | 8.31 | 7.58 | 27.95 | 27.08 | 42.99 | 33.95 | | | | | | |
| | PCA | 0.22 | 0.14 | 0.16 | 0.14 | 7.88 | 7.32 | 8.31 | 7.57 | 32.31 | 28.06 | 42.7 | 29.74 | | | | | | |
| | POS | 0.26 | 0.14 | 0.18 | 0.15 | 7.75 | 7.18 | 8.35 | 7.53 | 10.09 | 26.23 | 29.84 | 22.73 | | | | | | |
| Resting | CHROM | 0.36 | 0.13 | 0.39 | 0.16 | 6.34 | 8.38 | 6.71 | 8.11 | 1.75 | 49.09 | 3.34 | 35.94 | 0.99 | 0.97 | 0.93 | 75.86 | 37.56 | 34.93 |
| | GREEN | 0.36 | 0.14 | 0.39 | 0.18 | 6.09 | 8.31 | 6.74 | 8.08 | 1.82 | 48.97 | 2.1 | 19.65 | | | | | | |
| | ICA | 0.34 | 0.16 | 0.37 | 0.18 | 6.26 | 8.14 | 6.78 | 8.07 | 10.71 | 23.68 | 9.53 | 33.12 | | | | | | |
| | LGI | 0.36 | 0.14 | 0.39 | 0.17 | 6.16 | 8.35 | 6.87 | 8.11 | 1.7 | 49.04 | 4.32 | 30.71 | | | | | | |
| | OMIT | 0.34 | 0.15 | 0.36 | 0.18 | 6.17 | 8.23 | 6.76 | 8.09 | 1.68 | 49.04 | 4.33 | 31.69 | | | | | | |
| | PBV | 0.34 | 0.15 | 0.36 | 0.18 | 6.25 | 8.18 | 6.78 | 8.09 | 5.2 | 45.18 | 8.85 | 30.24 | | | | | | |
| | PCA | 0.34 | 0.15 | 0.37 | 0.18 | 6.18 | 8.22 | 6.76 | 8.1 | 1.57 | 49.21 | 4.62 | 33.97 | | | | | | |
| | POS | 0.35 | 0.14 | 0.38 | 0.17 | 6.43 | 8.26 | 7.02 | 8.1 | 2.01 | 49.27 | 3.86 | 30.39 | | | | | | |
| Rotation | CHROM | 0.33 | 0.12 | 0.32 | 0.14 | 7.08 | 8.53 | 7.37 | 8.1 | 4.47 | 50.14 | 6.57 | 33.71 | 0.99 | 0.97 | 0.89 | 71.36 | 37.28 | 34.23 |
| | GREEN | 0.31 | 0.13 | 0.3 | 0.15 | 7.41 | 8.48 | 7.83 | 8.32 | 12.08 | 43.77 | 13.21 | 23.68 | | | | | | |
| | ICA | 0.3 | 0.13 | 0.29 | 0.15 | 7.51 | 8.46 | 7.86 | 8.31 | 13.87 | 33.01 | 14 | 34.17 | | | | | | |
| | LGI | 0.33 | 0.12 | 0.31 | 0.14 | 7.08 | 8.46 | 7.57 | 8.18 | 4.16 | 49.62 | 5.99 | 29.35 | | | | | | |
| | OMIT | 0.3 | 0.13 | 0.29 | 0.15 | 7.46 | 8.51 | 7.84 | 8.3 | 4.65 | 49.7 | 5.88 | 30.57 | | | | | | |
| | PBV | 0.29 | 0.13 | 0.28 | 0.15 | 7.53 | 8.49 | 7.88 | 8.31 | 11.39 | 37.07 | 11.78 | 29.7 | | | | | | |
| | PCA | 0.3 | 0.13 | 0.29 | 0.14 | 7.46 | 8.49 | 7.83 | 8.31 | 4.4 | 50.04 | 6.51 | 34.09 | | | | | | |
| | POS | 0.33 | 0.12 | 0.32 | 0.14 | 7.12 | 8.38 | 7.57 | 8.13 | 4.04 | 50.19 | 6.41 | 28.83 | | | | | | |
| Talk | CHROM | 0.21 | 0.08 | 0.18 | 0.08 | 7.69 | 8.56 | 7.83 | 8.24 | 11.39 | 47.11 | 21.96 | 30.01 | 0.96 | 0.98 | 0.84 | 63.3 | 36.52 | 32.55 |
| | GREEN | 0.19 | 0.09 | 0.17 | 0.1 | 8.01 | 8.42 | 8.08 | 8.21 | 21.36 | 41.73 | 21.09 | 29.04 | | | | | | |
| | ICA | 0.19 | 0.1 | 0.16 | 0.1 | 8.01 | 8.35 | 8.08 | 8.22 | 17.47 | 33.22 | 21.54 | 35.9 | | | | | | |
| | LGI | 0.21 | 0.08 | 0.17 | 0.09 | 7.74 | 8.49 | 7.82 | 8.21 | 12.65 | 47.34 | 21.32 | 30.15 | | | | | | |
| | OMIT | 0.19 | 0.1 | 0.16 | 0.09 | 7.93 | 8.41 | 8.01 | 8.21 | 12.55 | 47.37 | 21.27 | 30.2 | | | | | | |
| | PBV | 0.19 | 0.1 | 0.16 | 0.1 | 7.97 | 8.36 | 8.05 | 8.21 | 17.08 | 35.49 | 22.32 | 43.78 | | | | | | |
| | PCA | 0.19 | 0.09 | 0.17 | 0.1 | 7.95 | 8.4 | 8.03 | 8.21 | 11.7 | 45.69 | 20.01 | 30.42 | | | | | | |
| | POS | 0.21 | 0.08 | 0.17 | 0.09 | 7.79 | 8.36 | 7.86 | 8.23 | 8.45 | 47.49 | 21.27 | 31.74 | | | | | | |

The value of the SSIM and PSNR of video modified by our method, by Chen's method[25], and by projected gradient descent (PGD) attack[26], grouped by video activity.
*BPM* beats-per-minute, *DTW* dynamic time warping, *PPG* photoplethysmography, *rPPG* remote photoplethysmography, *|r|* correlation, *SSIM* structural similarity index measure, *PSNR* peak-signal-to-noise ratio.

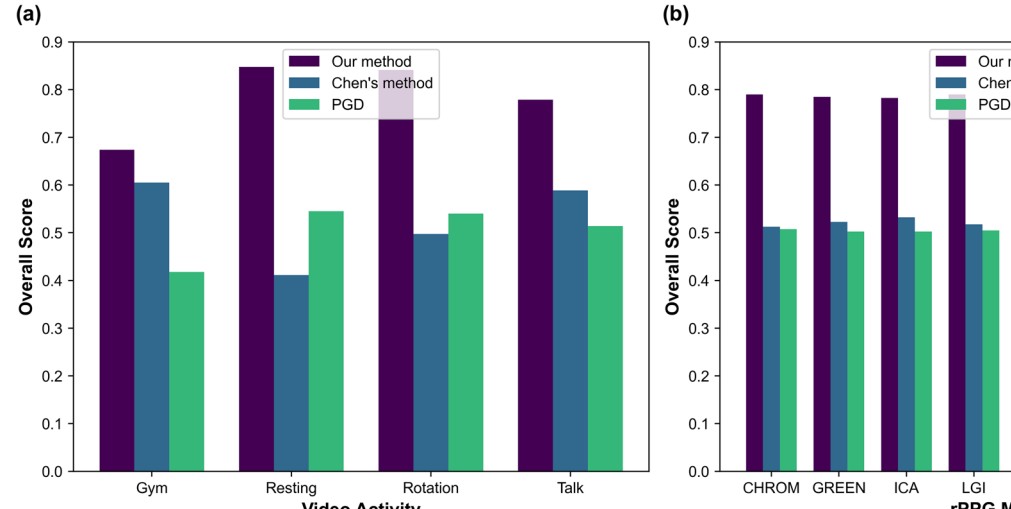

**Fig. 6 | Comparison of average overall scores between our method, Chen's method[25], and the projected gradient descent (PGD) attack[26]. a** Performance comparison across different physical activities (gym, resting, rotation, and talking), averaged over all rPPG techniques (*n* = 8) and all subjects (*n* = 6) over the entire length of the video. **b** Performance comparison across different rPPG measurement techniques, averaged over all activities (*n* = 4) and all subjects (*n* = 6) over the entire length of the video. Source data provided in Table 1.

and talking activities, while our method performs better during resting and rotation.

Our method achieves a consistently lower correlation coefficient between the rPPG signals extracted from the modified videos and the ground truth across all activities and rPPG measurement techniques, outperforming both Chen and PGD. On average, our method achieves a correlation coefficient of 0.125, compared to 0.252 for Chen and 0.14 for PGD. This demonstrates that our approach is more effective at disrupting the rPPG signal, providing enhanced privacy protection.

Is another metric for assessing the similarity between the modified rPPG signal and the ground truth, where a higher DTW distance indicates greater dissimilarity between the signals. Our method achieves a lower DTW for the gym activity but slightly higher DTW for other activities. Overall, our method achieves the highest average DTW of 8.10, compared to Chen's method at 7.69, and PGD at 8.01.

In terms of heart rate (HR) estimation error from the extracted rPPG signal, our method induces substantially more error across all activities and rPPG techniques, averaging a |ΔBPM| of 22.24 bpm higher than Chen and 10.08 bpm higher than PGD. For activities such as resting, rotation, and talking, our method outperforms both Chen and PGD by a substantial margin. Specifically, for resting, our algorithm introduces 40.31 bpm more error than Chen and 14.72 bpm more than PGD. Similarly, for rotation, our algorithm outperforms Chen by 36.65 bpm and PGD by 14.93 bpm. In the talking activity, our method performs better by 21.84 bpm and 10.52 bpm, respectively. However, for the gym activity, our method performs comparably to PGD with only a negligible difference, but it underperforms compared to Chen's method by 9.83 bpm.

**Overall score**

Figure 6a compares the average overall score over all participants, windows, and rPPG methods across video activity on the video modified by Chen's method and the video modified by our method. We could see that our method performed better than Chen's method over all video activities. Figure 6b compares the average overall score over all participants and windows across the rPPG method on the video modified by Chen's method and the video modified by our method. We could see that our method performed better than Chen's method over all rPPG methods. And our method performs uniformly well across different rPPG methods using different evaluation metrics, as shown in Fig. 7. Table 1 presents the

numerical values for the average overall score grouped by video activities and rPPG methods.

## Discussion

Methods such as Privacy-Phys[26], employing PGD via pre-trained 3DCNN, and Hsieh et al.'s approach[24] using double-cycle consistent learning share a common constraint–they require fixed-size frame inputs. This necessitates frame resizing or cropping when dealing with differing frame sizes, potentially resulting in detail loss and degraded visual quality, notably for images with larger facial areas than the networks' input size. In contrast, our approach accommodates all frame sizes, eliminating the need for adjustments and preserving original content integrity.

Additionally, previous methods primarily rely on window-wise processing, which requires access to partial or entire video segments for tasks such as optimization[23,26], principal component decomposition[25], or input into deep learning networks[24,27]. This approach introduces processing delays, limiting the applicability of these methods in real-time scenarios. In contrast, our video-editing model operates on a frame-wise basis, requiring only a single frame at a time, making it considerably faster and more suitable for real-time applications such as video calls.

When compared to Chen's method[25], which requires processing multiple frames or full video windows, our approach operates on individual frames and does not rely on temporal aggregation. Similarly, unlike Privacy-Phys[26] and related techniques that use large 3D CNNs for signal manipulation, our method utilizes a lightweight MediaPipe model to extract a binary mask, followed by a simple sinusoidal modulation within this region. While we do not provide direct runtime benchmarks, this streamlined design suggests a lower computational footprint, making it amenable to deployment on resource-constrained hardware. Overall, our framework presents a frugal and potentially effective alternative for secure, low-latency rPPG signal modification and transmission, with practical advantages in scenarios where minimal overhead is desirable.

Furthermore, our approach maintains consistent performance across varying video activities, overcoming the limitations seen in methods like Chen's[25] and double-cycle consistency learning[24], which exhibit uneven modification effects. We advocate for result grouping by video activities and rPPG methods for nuanced performance evaluation.

Our method stands out by addressing the limitations and inconsistencies inherent in previous models and offering a comprehensive

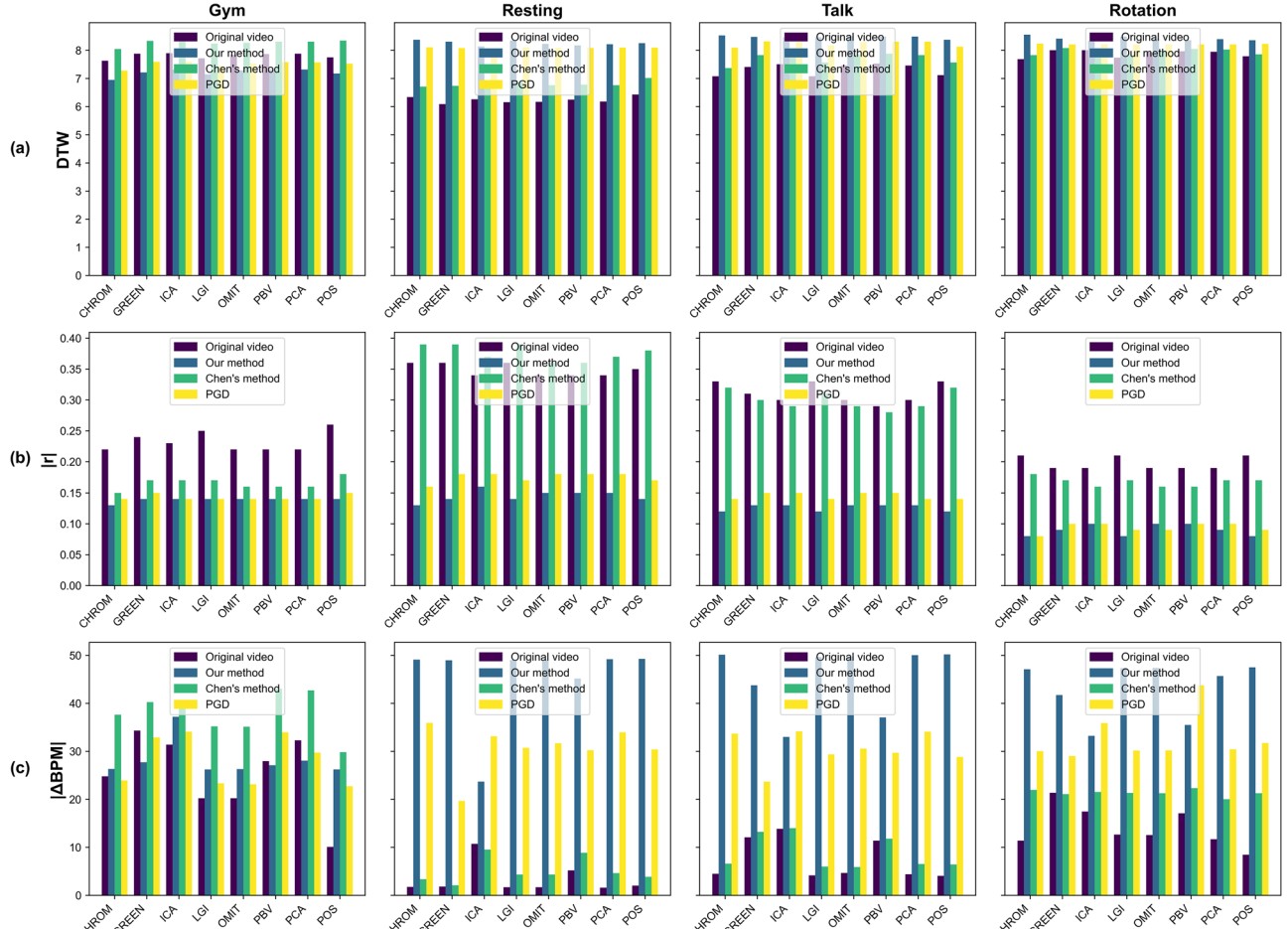

**Fig. 7 | Performance comparison of the different video modification algorithms.** Bar plots showing the average performance across participants ($n = 6$) for each rPPG estimation method applied to four types of input videos: original video (red), our modified video (orange), video modified by Chen's method (green), and video modified using a projected gradient descent (PGD) attack (blue). Metrics are grouped by video activity (Gym, Resting, Talk, and Rotation) and include: **a** DTW, **b** $|r|$, and **c** $|\Delta BPM|$. Each bar represents the average value over all windows for all participants. Note that BPM beats-per-minute, DTW dynamic time warping, PPG photoplethysmography, rPPG remote photoplethysmography, $|r|$ correlation. Source data provided in Table 1.

solution that synergizes technical proficiency with practical applicability, paving the way for future innovations in the domain. It offers lightweight modification capabilities and versatile applicability and upholds the visual fidelity of the original content, thus presenting a balanced and refined solution in the evolving landscape of video modification and secure transfer technologies.

Moreover, we provide a complete framework for modifying video to hide the rPPG signal to securely transfer to recover the original video.

Below is a summary of the recommendations that have been formulated based on the findings of this study. We suggest that instead of comparing the mean absolute error of the estimated heart rate from the modified video and the reference heart rate, researchers should compare the correlation and DTW of the rPPG signal with the reference PPG signal, since heart rate is derived from the rPPG signal. We also advise using an overall evaluation score that balances modification effectiveness and resulting video fidelity when comparing the relative performance of different rPPG modification methods. In addition, we recommend assessing the performance of rPPG modification methods across different video activities and rPPG methods, since a method may appear effective on average but perform poorly under specific conditions. Furthermore, we suggest that the complexity and running time of rPPG modification methods should be considered for real-time applications. Extending the modulation approach to the red and blue channels may further enhance privacy protection by ensuring that no physiological information is leaked through these channels. Finally, we recommend evaluating rPPG

modification methods on additional datasets or combinations of datasets to further validate robustness and generalizability across diverse real-world scenarios.

In conclusion, our innovative approach offers a simple, rapid, and flexible method for concealing original rPPG signals in facial videos by introducing a sinusoidal signal to the green channel over time. Unlike previous techniques, which necessitate fixed-size frames and involve window-wise processing that causes delays, our method accommodates varying frame sizes without cropping or resizing and enables instantaneous frame-wise processing. While we did not include direct runtime benchmarks, the algorithm's simplicity and independence from temporal context suggest suitability for low-latency applications, which may support real-time use cases such as online meetings.

Additionally, our method demonstrates uniform efficacy across different video activities, overcoming the inconsistent performance observed in other methods, such as Chen's method and double-cycle consistency learning, particularly with resting subjects and subjects exhibiting head movements, respectively. Moreover, we introduce a comprehensive framework spanning from modifying video to conceal rPPG signals to securely transferring and eventually recovering the original video, ensuring that only the intended receiver can regain the original rPPG signal.

## Data availability
The LGI-PPGI dataset analyzed in this work is publicly available under a CC-BY-4.0 license at https://github.com/partofthestars/LGI-PPGI-DB.

Source data underlying all figures and tables in this manuscript are provided directly within the manuscript (see Table 1).

## Code availability

All code used in this study is available at https://github.com/saksham2001/Balancing-Cardiac-Privacy. An archived, citable snapshot is available at Zenodo[42].

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

## Acknowledgements
This work was funded by Khalifa University (grant number FSU-2025-001).

## Author contributions
M.E. designed and led the study. A.Y., M.E., S.B., and C.M. jointly conceived the study. M.E., A.Y., and S.B. implemented the methodology and wrote the initial draft of the manuscript. All authors have read and agreed to the published version of the manuscript.

## Funding

## Competing interests
The authors declare no competing interests.
