## [Transparent Peer Review file · Communications Medicine]

Balancing Cardiac Privacy with Quality in Video Recordings

Corresponding Author: Professor Mohamed Elgendi

Version 0:

Reviewer comments:

Reviewer #1

(Remarks to the Author)

[Paper Summary] This paper proposes a technique for removing rPPG signals from RGB face videos to conceal a person's physiological information. To do this, a system is constructed to safely transport videos with modified physiological information to a public domain with a key being provided to the intended receiver to decrypt real physiological information present in the video. Concealing the physiological characteristics present in a video is done by adding a sinusoidal signal to the skin pixels in the green channel of a face video. The authors propose to combine multiple video and physiological quality metrics into one to verify the overall quality of the videos they process. Experimental results are provided using the LGI-PPGI dataset to analyze the tradeoff between video quality and physiological privacy.

[Overall comments]

Given a good amount of work being done on this topic, the current paper's contribution is relatively minor and incremental. It is not clear how justified when they consider some recent techniques using iterative updates/refinement as time-consuming, as the paper mainly compares with an earlier work rather than the more recent work they briefly cited and commented on in the early part of the paper. There is little critical analysis of the limitations of the proposed strategy – which basically uses sinusoids to a color channel to disguise the original heart rate rather than removal, suggesting a motivated adversary may circumvent the privacy protection. There are also major grammatical issues and many typos.

Detailed comments

[Strengths]

- 1) The authors introduce a framework for addressing an important privacy concern associated with capturing facial videos of people.
- 2) Experimental results include a variety of different rPPG algorithms for comparison.
- 3) The results suggest that the high visual quality of videos can be maintained after they have been modified by the proposed approach.

[Technical Questions/Issues]

1. The method imposes a sinusoidal signal onto the face video, but how do we know if it removes the true rPPG signal?
2. The modulation described in the proposed algorithm is only applied to the green channel, but the red and blue channels may also have physio information in them. Can't this also lead to privacy leakage?
3. If Sun's method [21] has dependency issues, as stated by the authors, then why not select another algorithm for comparison?
4. The authors mention these statements in their introduction to explain the drawbacks of other approaches:
 - a) "However, all of the methods either involve time consuming processing steps such as optimization and principal component analysis, or create visible artifacts."
 - b) A similar statement is reiterated in "This approach demands the processing of partial to entire video windows for

optimization [18, 21], principal component decomposition [20], or network feeding [19, 22], introducing delays and narrowing their application scope. Our method, utilizing a fast, frame-wise process, suits common scenarios like online meetings, allowing for synchronous frame modification and display to inhibit unauthorized rPPG signal detection.”

The authors should provide runtime analysis to support such claims. For example, even though Chen et al. [18] use optimization, their runtimes are still quite efficient with the most expensive module still running at 10 fps. This is close to real-time and may have room for further speed-up with optimization in software implementation.

[Weaknesses] By Sections.

Grammar:

This manuscript needs MAJOR grammatical revision and careful proofread to show a high-quality effort. I've provided some examples of grammatical issues below., but going through the entire manuscript would be too time-consuming. The authors may consider using grammar-checking tools such as Grammarly, etc.

1. Mis-spellings (e.g. singals instead of signals in the title of their second section), improper noun-verb agreement, etc.
2. “While mobile computing technologies enables” (<--- should be “enable”)
3. “...it also arises security and privacy challenges” (<— not good grammar)
4. “ As defined by National Committee for Vital and Health Statistics (NCVHS)” (<--- Put “the” in front of National Committee)

Introduction:

1. This section should be made sharper.

1) A more concrete layout of the contributions should be presented in this section.

2) A description of the proposed algorithm for modifying rPPG is missing in the introduction and should be clearly stated. While the authors mention some drawbacks of other works, they don't mention how their work addresses these drawbacks. Clearly stating this will allow readers to understand exactly what privacy issues will be addressed and in what situations their framework is and is not applicable. This will prepare the reader for what is to come in the body of the paper.

2. A clearer outline of the problem setup, with all of its constraints laid out, is recommended. While the general idea can be understood after reading the paper, some of the details are scattered. For example, in the problem description in the introduction section, the authors mention nothing about a sender or receiver and do not mention any motivation for encryption. They only mention that it is desired to have a video in which a person's physiological information is removed. Yet, this information is included in their algorithmic pipeline.

Algorithms to modify the rPPG signals:

1. The block diagram overview is a bit disorganized. The fonts are small and it does not flow naturally. Restructuring this figure to improve its quality and also take better advantage of empty space in the modules is recommended.

2. The overall algorithmic contribution is not very novel. The video editing model works by simply adding a sinusoid to green channel skin pixels. This does not mean that the heart rate trace in the frequency domain is removed. It just means that a new frequency trace is added. Moreover, there is variability in a person's natural PPG signal (this is called heart rate variability). Could simply adding a sinusoid of constant frequency signal to an attacker that the heart rate added to the video is fake?

Evaluation Pipeline:

The evaluation pipeline creates a metric for measuring the balance between video quality and modification effectiveness. However, the measures under modification effectiveness do not necessarily capture the removal of the true heart rate in the frequency domain. For example, imposing a new, stronger, periodic signal on the skin pixels can fool this metric into thinking the heart rate has been removed, but when in reality, it just happens to be the case that 2 frequency traces appear in the frequency domain (the second strongest being the HR trace). This may favor the authors' algorithm.

Results:

1. Fonts in figures and plots (axes, labels, etc.) are too small and should be improved

2. The authors provide minimal discussion of the results they provide in this paper (one small paragraph is used to describe 3 figures and 1 table). More insights should be provided to explain the results. Why do the authors suspect that their method achieves superior performance over others? When does it work well and when does it fail? What are the limitations of the proposed algorithm?

3. Only one dataset is provided for analysis in this paper.

4. Neural network methods like [21] typically work best in situations in which the training and testing datasets contain similar environmental conditions (lighting conditions, skin tone diversity, etc.). It is recommended that the authors comment on this issue. Are the environmental conditions that were used for training [21] similar to those in the PURE dataset or not?

5. Providing spectrograms of the rPPG signals extracted from the modified face videos would be useful to the audience to see the removal of the original heart rate trace.

Reviewer #2

(Remarks to the Author)

This paper proposes a new method to modify rPPG signals in facial videos. Different from previous works that only focus on modifying or removing original rPPG signals, the method can conceal the original rPPG signals during video transferring and recover the original rPPG signals when receiving the videos. Results are also compared with previous rPPG modification methods and show performance advantages. Here are some suggestions for minor revisions.

1. Please add the computational speed in the result part.

2. Please add a figure for the video editing model about how to insert the sinusoidal signals into videos. (e.g., facial areas, different segments have different inserted sinusoidal signals)

3. Figure 1 should have higher resolution.

4. The format of the caption of figure 4 should be fixed.

5. There are some grammar errors in the paper, please check them.

6. It is better to compare the proposed method to other related methods using multiple commonly used metrics, which have specific and well-understood meanings, rather than the 'overall score'.

Version 1:

Reviewer comments:

Reviewer #2

(Remarks to the Author)

About my first comment on computational speed:

It is not appropriate for the authors to claim 'efficiency' or 'real time' since no evaluation results are provided in any form. If the authors do want to claim the advantages of the method's computational efficiency over other methods, experimental comparisons are needed. Otherwise, the corresponding claims should be removed.

The other questions/comments have been addressed.

Reviewer #4

(Remarks to the Author)

The authors have answered all the comments sincerely but in the abstract section, specific novelty of the method and an outline of the applied methodology should be mentioned properly. Otherwise, the manuscript is properly revised.

Reviewer #1 (Remarks to the Author):

Technical Questions/Issues:

1. **The method imposes a sinusoidal signal onto the face video, but how do we know if it removes the true rPPG signal?**

Author Response: Thank you for your question. The method aims to obscure the true rPPG signal by superimposing a sinusoidal signal, ensuring that the original cardiac information is masked while preserving video quality. The effectiveness of this approach is validated through two complementary metrics: the correlation coefficient, which measures the strength of similarity between the extracted rPPG signal from the modified video and the original PPG signal, and Dynamic Time Warping (DTW), which evaluates the temporal alignment between the two signals. A significant reduction in these metrics indicates that the modified video no longer retains meaningful cardiac information, providing strong evidence that the true rPPG signal has been successfully masked. These metrics are widely used and validated in rPPG literature for assessing physiological signal similarity and temporal alignment.

Author Action: We added the following text to the manuscript in the "Evaluation Pipeline" section:

“We validate the effectiveness of the proposed method by calculating the correlation coefficient and Dynamic Time Warping (DTW) between the modified rPPG signal and the original PPG signal. These metrics assess both temporal alignment and signal similarity, which are critical for evaluating whether the true rPPG signal has been successfully removed or masked. Many studies have shown that metrics like the correlation coefficient³² and DTW^{26,33} are effective in comparing PPG signals. A significant reduction in these metrics indicates successful removal or masking of the true rPPG signal.”

2. **The modulation described in the proposed algorithm is only applied to the green channel, but the red and blue channels may also have physio information in them. Can't this also lead to privacy leakage?**

Author Response: We focused on modulating the green channel because prior studies in rPPG estimation have consistently shown that the green channel carries the most significant rPPG information due to the high absorption of hemoglobin in this wavelength range. While the red and blue channels do contain some physiological signals, their contribution is much smaller and less reliable for rPPG extraction. Modifying these channels would add minimal privacy benefit but could degrade the perceptual quality of the video and increase computational complexity, impacting real-time performance. Therefore, our method makes a deliberate trade-off: focusing on the green channel to ensure efficient signal modification while maintaining high video quality and processing speed. We believe this strikes a reasonable balance between privacy and performance.

Author Action 1: We have added the following text to better explain the reasoning for solely using the green channel in the "Video Editing Model" section under "Algorithms to modify the rPPG signals":

"... Moreover, multiple studies^{25,26} have shown that the green channel carries the most significant rPPG information, owing to the peak absorption of hemoglobin at green wavelengths. While the red and blue channels contain minor physiological information, they are less reliable for heart rate extraction and contribute more noise. Thus, we focus our modifications solely on the green channel to maximize the masking of rPPG information while minimizing video distortion. Limiting the modification to the green channel also ensures lower computational overhead. ..."

Author Action 2: We have also added the following recommendation for future researchers in the "Discussion" section:

"We suggest extending the modulation approach to the red and blue channels to further enhance privacy protection by ensuring that no physiological information is leaked through these channels."

3. **If Sun's method [21] has dependency issues, as stated by the authors, then why not select another algorithm for comparison?**

Author Response: Thank you for your valuable feedback. The privacy problem in rPPG is still an emerging topic with limited published literature. The current body of work in this area can be broadly categorized into two approaches: (1) classical signal processing methods for removing physiological signals, and (2) deep learning-based optimization techniques that aim to maximize errors in heart rate estimation while preserving video fidelity. We selected Chen et al.'s method as it represents the classical signal processing approach, and Sun et al.'s method as it is one of the few papers employing deep learning for this problem. Notably, Sun et al.'s method is a rare example of applying a deep learning-based optimization framework for rPPG privacy.

However, the authors of Sun et al.'s method did not make the model weights or detailed implementation available, specifically for the 3DCNN used in their work. As a result, we were unable to replicate the method exactly and opted to use a pre-trained EfficientPhys model, acknowledging that this may not fully reflect the original implementation. We explicitly mentioned this limitation in the manuscript.

Author Action: We updated the text in "Previous Methods" sub-section to provide some reasoning as to why these baselines were chosen.

"For comparison, we selected two representative methods from the literature. First, we re-implemented Chen *et. al.*'s method²⁰, which removes rPPG signals by applying a soft elimination factor to the principal component with the largest normalized band power.

This method serves as an example of a classical signal-processing approach. Second, we attempted to re-implement Sun et al.'s method²¹, which relies on a pre-trained 3DCNN-based optimization framework, similar to a projected gradient descent attack. However, Sun *et. al.*'s 3DCNN model and training details were not publicly available. Due to this, we used EfficientPhys²³, a pre-trained model on the PURE dataset²⁴, as a substitute for Sun *et. al.*'s 3DCNN. While this allowed us to compare deep learning approaches, we acknowledge that this substitution may not fully reflect the performance of Sun *et. al.*'s original implementation.”

4. **The authors should provide runtime analysis to support claims about efficiency.**

Author Response: Thank you for this suggestion. While we acknowledge the importance of demonstrating efficiency, a direct runtime comparison with previous methods is challenging due to fundamental differences in processing approaches. Our work presents an end-to-end framework that includes signal modification, encryption, transmission, decryption, and signal restoration, which is more comprehensive than existing individual methods focused solely on modification.

Comparing our method with Chen's approach is particularly difficult as their method requires processing multiple frames or an entire video segment to perform principal component decomposition. In contrast, our method operates on a frame-by-frame basis, modifying each frame independently and thus eliminating the need for extensive temporal data, which inherently makes it more suitable and faster for real-time applications.

Our approach also offers significant speed advantages over Privacy-Phys and similar deep learning-based methods. Privacy-Phys relies on a large 3D CNN model for signal modification, which is computationally expensive and requires substantial processing power. Our method, on the other hand, uses a lightweight MediaPipe model for binary mask extraction and a simple sinusoidal modulation, making it far more resource-efficient and capable of running on simpler hardware without the need for deep learning models.

Author Action: We have added a detailed explanation of our method's advantages in terms of runtime efficiency in the "Discussion" section. The following text has been added to provide clarity on the merits of our approach:

“Additionally, previous methods primarily rely on window-wise processing, which requires access to partial or entire video segments for tasks such as optimization^{18,21}, principal component decomposition²⁰, or input into deep learning networks^{19,22}. This approach introduces processing delays, limiting the applicability of these methods in real-time scenarios. In contrast, our video editing model operates on a frame-wise basis, requiring only a single frame at a time, making it significantly faster and more suitable for real-time applications such as video calls.

When compared to Chen's method²⁰, which necessitates processing multiple frames or entire video windows, our approach is inherently more efficient for real-time use. Additionally, unlike Privacy-Phys²¹ and similar methods that employ large 3D CNN models for signal modification, our method leverages a lightweight MediaPipe model to extract a binary mask, followed by a simple sinusoidal modulation within this mask. This streamlined design is highly resource-efficient, allowing our algorithm to run on simpler hardware while maintaining effectiveness. Overall, our framework offers a frugal and robust solution for secure, real-time rPPG signal modification and transmission, providing clear advantages over prior methods in terms of processing speed, computational efficiency, and real-time applicability."

Grammar and Clarity:

Grammar Issues:

- "singals" instead of "signals"
- "enables" instead of "enable"
- "...it also arises security and privacy challenges" should be "...it also raises security and privacy challenges" wza3\
- "As defined by National Committee for Vital and Health Statistics (NCVHS)" should be "As defined by the National Committee for Vital and Health Statistics (NCVHS)"

Author Response: Thank you for highlighting these mistakes. We have used grammar checking tools and have promptly fixed spelling and grammar errors throughout the manuscript.

Author Action: We corrected spelling and grammar issues throughout the manuscript.

Introduction:

1. More concrete layout of contributions and a description of the proposed algorithm:

Author Response: Thank you for your valuable suggestion. We agree that clarifying the contributions and providing a detailed description of the proposed algorithm will improve the clarity of the manuscript. Based on your feedback, we have made significant changes to the introduction section.

Author Action 1: We have added the following text to provide a clearer description of the proposed algorithm:

“To address the limitations of previous methods, we propose a framework which includes a sender who modifies the video to obscure the rPPG signals and a receiver who decrypts and recovers the original video using a secure key. To the best of our knowledge, this is the first reversible method that allows the physiological signals to be reintroduced into the video. In our approach, we

generate a list of frequencies using RSA encryption. A sinusoidal signal, based on these frequencies, is added frame-wise to the green channel of the video's time series. Along with the modified video, the encrypted frequency list is transmitted to the receiver, who can then decrypt the list and reverse the modification, restoring the original physiological signal. This setup ensures that anyone with access to the video cannot view or extract the original rPPG information, nor can they discern how the video was altered, as the frequency list remains securely encrypted."

Author Action 2: We have also revised the contributions section to explicitly highlight the key contributions of our work:

"To summarize, our contributions include the following:

- We propose the first end-to-end reversible algorithm for removing, encrypting, transmitting, and restoring rPPG signals in facial videos. Our method operates in real-time, frame-wise, without relying on explicit using rPPG signal, making it faster than prior approaches.
- We introduce a novel evaluation metric that compares rPPG signals using correlation (r) and dynamic time warping (DTW), and propose an overall score that balances video fidelity and modification effectiveness.
- We perform an in-depth analysis by evaluating the generalization ability of the rPPG modification method across different activities, identifying both strengths and potential limitations."

2. Clearer outline of problem setup:

Author Response: Thank you for pointing out the need for a clearer problem setup. We have revised the introduction accordingly.

Author Action 1: We have added the following text to introduce the motivation and problem setup:

"However, the rise of rPPG techniques has also introduced significant privacy concerns. The presence of physiological signals in facial videos allows for the unauthorized extraction of sensitive health information. Through rPPG, it is possible to infer an individual's physiological state without their knowledge or consent. This raises the risk of exploitation, where such data could be leveraged for unethical purposes, such as covertly monitoring a person's health condition or emotional state in contexts like targeted marketing, negotiations, or financial transactions¹⁸. Consequently, there is an urgent need for methods that can selectively modify or obfuscate rPPG-related information in facial videos, ensuring the protection of vital signs while preserving overall video fidelity. In this work, we particularly focus on the real-time removal of physiological signals from facial videos transmitted in scenarios such as video calls, live streaming, and

remote patient monitoring—ensuring privacy protection without compromising the visual quality of the transmitted video.”

Author Action 2: We have also added the following text to explain the sender-receiver framework and algorithm:

“To address the limitations of previous methods, we propose a framework which includes a sender who modifies the video to obscure the rPPG signals and a receiver who decrypts and recovers the original video using a secure key. To the best of our knowledge, this is the first reversible method that allows the physiological signals to be reintroduced into the video. In our approach, we generate a list of frequencies using RSA encryption. A sinusoidal signal, based on these frequencies, is added frame-wise to the green channel of the video’s time series. Along with the modified video, the encrypted frequency list is transmitted to the receiver, who can then decrypt the list and reverse the modification, restoring the original physiological signal. This setup ensures that anyone with access to the video cannot view or extract the original rPPG information, nor can they discern how the video was altered, as the frequency list remains securely encrypted.”

Algorithms to Modify the rPPG Signals:

1. Restructuring block diagram overview:

Author Response: Thank you for highlighting this. We agree that the original Figure 1 was not structured clearly, which may have made it challenging to follow the overall workflow.

Author Action: We have updated Figure 1 in the manuscript to increase font sizes and reorganize the layout for better clarity. Additionally, we have broken down the original diagram and added a new figure (Figure 2) that specifically focuses on the video editing model, separating it from the broader pipeline overview.

2. Novelty of the algorithm:

Author Response: Thank you for the suggestion. We have emphasized the novelty of our frame-wise sinusoidal modulation approach and its effectiveness in maintaining video quality while obscuring rPPG signals.

Author Action: We have added the following text to the "Methods" section:

“... Our method is novel in its simplicity and efficiency, applying a frame-wise sinusoidal modulation to the green channel to effectively obscure rPPG signals without the need for complex optimization or training processes. Unlike other approaches that require the estimation of a time-series signal from multiple

frames, our method operates on a single frame, eliminating the dependency on temporal information and reducing computational complexity.”

Evaluation Pipeline:

1. Detailed explanation of results:

Author Response: Thank you for your suggestion. We have made significant improvements to the clarity and detail in the results section.

Author Action: We have added the following text to provide a more thorough explanation of the results:

“Table 2 presents the evaluation metrics for our method, Chen’s method²⁰, and the projected gradient descent (PGD) attack²¹, separated by four different activities—gym, resting, rotation, and talking—as well as different rPPG measurement techniques. **PSNR.** The PSNR column in Table 2 shows the average PSNR between the modified video and the original video across activities. Our method consistently achieves significantly higher PSNR values compared to both Chen’s method and PGD, approximately doubling their performance across all activities. Specifically, our method reaches an average PSNR of 67.87, compared to 37.02 for Chen and 33.48 for PGD. In employing our method, we utilize the lossless video coding format, FFV1.

Consequently, the frames wherein $\sin(0)$ is added to the green channel remain identical to the original frames, causing the PSNR to approach infinity. To facilitate numerical comparison, we have capped the maximum PSNR at 100 instead of allowing it to reach infinity.

SSIM. As shown in Table 2, our method consistently outperforms PGD by approximately 0.1 SSIM across all activities. The difference between our method and Chen’s method is negligible, with both achieving an average SSIM of 0.97. Notably, Chen’s method performs slightly better in the gym and talking activities, while our method performs better during resting and rotation.

Correlation Coefficient. Our method achieves a consistently lower correlation coefficient between the rPPG signals extracted from the modified videos and the ground truth across all activities and rPPG measurement techniques, outperforming both Chen and PGD. On average, our method achieves a correlation coefficient of 0.125, compared to 0.252 for Chen and 0.14 for PGD. This demonstrates that our approach is more effective at disrupting the rPPG signal, providing enhanced privacy protection.

DTW is another metric for assessing the similarity between the modified rPPG signal and the ground truth, where a higher DTW distance indicates greater dissimilarity between the signals. Our method achieves a lower DTW for the gym

activity but slightly higher DTW for other activities. Overall, our method achieves highest average DTW of 8.10, compared to Chen's method at 7.69, and PGD at 8.01.

|\DeltaBPM|. In terms of heart rate (HR) estimation error from the extracted rPPG signal, our method induces significantly more error across all activities and rPPG techniques, averaging a $|\Delta\text{BPM}|$ of 22.24 bpm higher than Chen and 10.08 bpm higher than PGD. For activities such as resting, rotation, and talking, our method outperforms both Chen and PGD by a substantial margin. Specifically, for resting, our algorithm introduces 40.31 bpm more error than Chen and 14.72 bpm more than PGD. Similarly, for rotation, our algorithm outperforms Chen by 36.65 bpm and PGD by 14.93 bpm. In the talking activity, our method performs better by 21.84 bpm and 10.52 bpm, respectively. However, for the gym activity, our method performs comparably to PGD with only a negligible difference, but it underperforms compared to Chen's method by 9.83 bpm.

Overall Score

Figure 6(a) compares the average overall score over all participants, windows, and rPPG methods across video activity on video modified by Chen's method and video modified by our method. We could see that our method performed better than Chen's method over all video activities. Figure 6(b) compares the average overall score over all participants and windows across the rPPG method on video modified by Chen's method and video modified by our method. We could see that our method performed better than Chen's method over all rPPG methods. And our method performs uniformly well across different rPPG methods using different evaluation metrics, as shown in Figure 7. Table 1 presents the numerical values for the average overall score grouped by video activities and rPPG methods.

2. Multiple datasets for analysis:

Author Response: Thank you for your suggestion. We selected the LGI PPGI dataset for our analysis because it provides a good balance with six subjects (five male, one female) performing four different activities. The dataset is recorded under both natural and artificial lighting conditions, making it a well-suited and widely used dataset in this area of research. Moreover, the dataset's diversity in activities and lighting conditions allows for a robust evaluation of rPPG signal modification algorithms. Due to computational resource constraints, we focused our analysis on this single dataset. The processing time and computational demand required for multiple large-scale datasets would have exceeded the scope of this study. However, we recognize the importance of generalizing our approach and plan to extend this work to multiple datasets in future studies.

Author Action: We have added the following recommendation to the "Discussion" section:

“We recommend evaluating rPPG modification methods on additional datasets or combination of datasets. This would further validate the robustness and generalizability of these methods across diverse real-world scenarios.”

Results:

1. Spectrograms of rPPG signals:

Author Response: Thank you for your suggestion. To address this, we have added spectrograms comparing the unmodified and modified rPPG signals. These spectrograms illustrate the effect of our modification method. Specifically, in the spectrogram of the unmodified video, the dominant frequency component aligns with the true physiological signal, as indicated by the estimated BPM line. In contrast, the spectrogram of the modified video shows additional frequency components around the injected frequency (100 bpm), demonstrating that our method successfully disrupts the original rPPG signal while maintaining the overall video quality.

Author Action: We have added a new figure (Figure 4) displaying the spectrograms of the unmodified and modified videos.

Reviewer #2 (Remarks to the Author):

1. Computational speed:

Author Response: Thank you for this suggestion. While we acknowledge the importance of demonstrating efficiency, a direct runtime comparison with previous methods is challenging due to fundamental differences in processing approaches. Our work presents an end-to-end framework that includes signal modification, encryption, transmission, decryption, and signal restoration, which is more comprehensive than existing individual methods focused solely on modification.

Comparing our method with Chen’s approach is particularly difficult as their method requires processing multiple frames or an entire video segment to perform principal component decomposition. In contrast, our method operates on a frame-by-frame basis, modifying each frame independently and thus eliminating the need for extensive temporal data, which inherently makes it more suitable and faster for real-time applications.

Our approach also offers significant speed advantages over Privacy-Phys and similar deep learning-based methods. Privacy-Phys relies on a large 3D CNN model for signal modification, which is computationally expensive and requires substantial processing power. Our method, on the other hand, uses a lightweight MediaPipe model for binary

mask extraction and a simple sinusoidal modulation, making it far more resource-efficient and capable of running on simpler hardware without the need for deep learning models.

Author Action: We have added a detailed explanation of our method’s advantages in terms of runtime efficiency in the "Discussion" section. The following text has been added to provide clarity on the merits of our approach:

“Additionally, previous methods primarily rely on window-wise processing, which requires access to partial or entire video segments for tasks such as optimization^{18,21}, principal component decomposition²⁰, or input into deep learning networks^{19,22}. This approach introduces processing delays, limiting the applicability of these methods in real-time scenarios. In contrast, our video editing model operates on a frame-wise basis, requiring only a single frame at a time, making it significantly faster and more suitable for real-time applications such as video calls.

When compared to Chen’s method²⁰, which necessitates processing multiple frames or entire video windows, our approach is inherently more efficient for real-time use. Additionally, unlike Privacy-Phys²¹ and similar methods that employ large 3D CNN models for signal modification, our method leverages a lightweight MediaPipe model to extract a binary mask, followed by a simple sinusoidal modulation within this mask. This streamlined design is highly resource-efficient, allowing our algorithm to run on simpler hardware while maintaining effectiveness. Overall, our framework offers a frugal and robust solution for secure, real-time rPPG signal modification and transmission, providing clear advantages over prior methods in terms of processing speed, computational efficiency, and real-time applicability.”

2. **Figure for video editing model:**

Author Response: Thank you for your valuable feedback. We acknowledge that the original Figure 1 did not clearly present the video editing model.

Author Action: We have created a new figure (Figure 2) dedicated to the video editing model, with more details about the editing model.

3. **Higher resolution for Figure 1:**

Author Response: Thank you for your suggestion. We agree that improving the resolution will enhance the figure’s clarity and readability.

Author Action: We have replaced Figure 1 with a higher-resolution version to ensure that all details are clearly visible.

4. **Caption of Figure 4:**

Author Response: Thank you for bringing this to our attention. We have fixed the caption and the overlapping problem by combining both the figures.

Author Action: We have combined Figure 4 and 5 (now figure 6). We have also updated the caption as follows.

“**Figure 6.** Comparison of average overall scores between our method, Chen’s method²⁰, and the projected gradient descent (PGD) attack²¹. (a) Performance comparison across different physical activities (gym, resting, rotation, and talking). (b) Performance comparison across different rPPG measurement techniques.”

5. Grammar errors:

Author Response: Thank you for bringing this to our attention. We have used grammar checking tools and have promptly fixed spelling and grammar errors throughout the manuscript.

Author Action: We have fixed language and grammar across the manuscript.

6. Comparison using multiple metrics:

Author Response: Thank you for your suggestion. In previous work on this topic, the standard metric for evaluating performance has been the mean absolute error (MAE) between the ground truth and the measured heart rate (in beats per minute), which quantifies the algorithm's effectiveness in removing the physiological signal. We have reported results using this metric. However, MAE in BPM focuses solely on heart rate, which is just one derivative of the broader rPPG signal. To provide a more comprehensive comparison, we have extended our analysis by including additional metrics that directly compare the full rPPG signals. Specifically, we use the correlation coefficient and dynamic time warping (DTW) distance, both of which offer a more detailed assessment of signal similarity. In terms of video fidelity, we evaluate the modified frames using SSIM and PSNR, which are widely accepted standards in the literature for measuring image quality. Furthermore, we introduce a new overall score that combines both rPPG signal accuracy and video fidelity metrics. This composite metric provides a more holistic measure of performance.

Author Action: To address this point in the manuscript, we have updated the text in the "Evaluation Pipeline" section as follows:

“We evaluate rPPG modification algorithms from two key perspectives: the effectiveness of removing the original rPPG-related information and the resultant video fidelity. Instead of limiting our analysis to derivatives like heart rate, we perform a more comprehensive evaluation by directly comparing the rPPG signals. This approach captures the broader impact of signal modification, as rPPG contains rich physiological information beyond heart rate alone. By

considering both signal accuracy and video quality, our evaluation provides a more thorough understanding of the trade-offs between privacy protection and visual fidelity. Figure 5 shows the evaluation pipeline.”

Reviewer #1 (Remarks to the Author):

1. About my first comment on computational speed:

It is not appropriate for the authors to claim ‘efficiency’ or ‘real time’ since no evaluation results are provided in any form. If the authors do want to claim the advantages of the method’s computational efficiency over other methods, experimental comparisons are needed. Otherwise, the corresponding claims should be removed. The other questions/comments have been addressed.

Author Response: We thank the reviewer for this thoughtful and constructive comment. We acknowledge that the manuscript did not include explicit runtime benchmarks or comparative computational evaluations. Based on this feedback, we have removed strong claims related to “efficiency” and “real-time performance.” Instead, we now use more measured phrasing such as “lightweight design” and “potential for low-latency applications” to accurately reflect the simplicity of our method without making unsupported performance claims. These changes have been made throughout the manuscript in accordance with the reviewer’s suggestion.

Author Action 1: In the Abstract, we updated the sentence to:

"Results from 24 videos indicate our method achieves an overall score above 0.75 across all rPPG methods, roughly 50% higher compared to previous methods, demonstrating effective physiological signal removal, superior video fidelity under diverse conditions, and a lightweight design."

Author Action 2: In the Previous Methods section, we revised the description of our approach to:

"Our method is novel in its simplicity and lightweight design, applying a frame-wise sinusoidal modulation to the green channel to effectively obscure rPPG signals without the need for complex optimization or training processes."

Author Action 3: In the Discussion, we updated the comparative analysis to:

"When compared to Chen’s method \cite{elimin}, which requires processing multiple frames or full video windows, our approach operates on individual frames and does not rely on temporal aggregation. Similarly, unlike Privacy-Phys \cite{privacyphys} and

related techniques that use large 3D CNNs for signal manipulation, our method utilizes a lightweight MediaPipe model to extract a binary mask, followed by a simple sinusoidal modulation within this region. While we do not provide direct runtime benchmarks, this streamlined design suggests a lower computational footprint, making it amenable to deployment on resource-constrained hardware. Overall, our framework presents a frugal and potentially effective alternative for secure, low-latency rPPG signal modification and transmission, with practical advantages in scenarios where minimal overhead is desirable."

Author Action 4: In the Conclusion, we modified the final paragraph to:

"Unlike previous techniques, which necessitate fixed-size frames and involve window-wise processing that causes delays, our method accommodates varying frame sizes without cropping or resizing and enables instantaneous frame-wise processing. While we did not include direct runtime benchmarks, the algorithm's simplicity and independence from temporal context suggest suitability for low-latency applications, which may support real-time use cases such as online meetings."

Author Action 5: In the Introduction, we removed unsupported claims of real-time performance in two locations. The revised sentences are:

"In this work, we particularly focus on the removal of physiological signals from facial videos transmitted in scenarios such as video calls, live streaming, and remote patient monitoring, ensuring privacy protection without compromising the visual quality of the transmitted video."

and

"Our method operates frame-wise, without relying on explicit use of the rPPG signal, simplifying the process compared to prior approaches."

Reviewer #2 (Remarks to the Author):

- 1. The authors have answered all the comments sincerely but in the abstract section, specific novelty of the method and an outline of the applied methodology should be mentioned properly. Otherwise, the manuscript is properly revised:**

Author Response: We thank the reviewer for this helpful and constructive comment. We agree that the original abstract did not adequately emphasize the key novelty of our work or provide sufficient detail on the methodology. Based on this feedback, we have revised the relevant sentence in the abstract to explicitly mention the end-to-end reversibility of our method, the use of sinusoidal modulation applied to rPPG-rich facial regions, and the core steps involved in the approach.

Author Action: In the Abstract, we updated the following sentence:

“This paper introduces the first end-to-end reversible algorithm for removing, encrypting, transmitting, and restoring rPPG signals in facial videos, using frame-wise sinusoidal modulation applied to specific rPPG-rich facial regions, with a focus on maintaining perceptual quality and concealing the true heart rate.”